# Factors Affecting Nurses’ Internal Transfer Intentions after the Introduction of COVID-19-Related Family Visiting Restrictions

**DOI:** 10.3390/healthcare10050959

**Published:** 2022-05-23

**Authors:** Yoshiko Kitamura, Hisao Nakai, Keiko Teranishi

**Affiliations:** School of Nursing, Kanazawa Medical University, 1-1 Uchinada, Kahoku 920-0265, Japan; kitamu@kanazawa-med.ac.jp (Y.K.); k-tera@kanazawa-med.ac.jp (K.T.)

**Keywords:** COVID-19, internal transfer intention, family visiting restrictions

## Abstract

Hospitals have established visiting restrictions to block coronavirus disease 2019 (COVID-19) external transmission routes. This study investigated factors associated with nurses’ internal transfer intentions and changes in their workloads, burdens, and daily lives owing to pandemic-related family visiting restrictions. Participants were nurses from three medical institutions designated for infectious diseases in Ishikawa Prefecture, Japan. An original self-report questionnaire was developed based on previous studies and a web-based survey conducted. Responses were received from 152 nurses and 84 were included in the analysis. Factors influencing internal transfer intentions were age ≥30 years [odds ratio (OR): 6.54, 95% confidence interval (CI): 1.19–35.83]; ≥11 years of experience (OR: 12.57, 95% CI: 2.32–68.02); and longer working hours (OR: 4.51, 95% CI: 1.48–13.72). The effect of visitation restrictions on daily life and internal transfer intentions was greater in nurses with ≥11 years of experience (OR: 4.31, 95% CI: 1.09–17.04), those with increased night awakening (OR: 3.68, 95% CI: 1.33–10.18), and those who desired to receive counseling (OR: 4.38, 95% CI: 1.07–17.91). In conclusion, excessive working hours may affect nurses’ internal transfer intentions during the COVID-19 pandemic. Nocturnal awakening and desire to receive counseling may predict nurses’ internal transfer intentions.

## 1. Introduction

Since the first COVID-19 case in November 2019 in Wuhan, China, the pandemic has continued to cause problems worldwide. As of 13 March 2022, the World Health Organization reported more than 455 million confirmed COVID-19 cases and 6 million deaths worldwide [1]. In Japan, the Omicron variant of COVID-19 has been spreading since the end of 2021, and the Japanese government has required citizens to refrain from attending large-scale events and to curtail activities such as outings and travel [2]. The number of new COVID-19 cases and the rate of hospital bed utilization are key indicators that determine the government’s advice to the public to refrain from regular activities. It is important to prevent the spread of COVID-19 within hospitals to maintain hospital functioning [3]. The Japanese Ministry of Health, Labor and Welfare has imposed hospital visiting restrictions to block external routes of virus transmission, and is recommending online visits. As of March 2022, visiting restrictions in hospitals and nursing homes were still in place [4]. Hospital visiting restrictions help to reduce the risk of external virus transmission. However, the lack of family support when explaining treatment and care plans to patients, absence of technical guidance and training for family members at discharge, and lack of encouragement and support for emotional distress through family visits are forcing nurses to change their methods of patient support and to adopt new measures [5].

According to a report on the effects of COVID-19 visiting restrictions, nursing homes have established local protocols for accepting visitors, which have had a positive effect on, and contributed to, the well-being of patients and their families. However, staff reported that the implementation of these protocols was stressful for both visitors and staff members because of the fear of infection risk and worries about their own health and the health of their families [5]. Nursing home physicians wish to provide individualized solutions in situations where visiting restrictions are necessary, but they face the added dilemma of assessing the dying phase of residents and making exceptions to these restrictions accordingly [6]. Families are very concerned about patient isolation and loneliness under nursing home visitation restrictions, and it has been suggested that providing information about patients’ psychological states may increase family satisfaction [7]. It has been reported that family members are more satisfied with telephone contact by nurses, contact from behind glass, and contact while maintaining physical distance than with video calls [8].

One study showed that the oral intake of patients with dementia worsened during periods when visits from close relatives were restricted [9]. Families of patients with severe acute brain injury, who have difficulty communicating, make decisions on their behalf. Considering the uncertainty of prognosis for these patients, families wish to be able to observe the patients’ appearances and receive regular and consistent information from healthcare teams [10].

Child–parent contact is essential in children’s hospitals. Healthcare teams at these hospitals repeatedly review case studies to determine how visiting restrictions are to be imposed and when exceptions are justified [11]. The effects on children and their parents of visitation restrictions in neonatal intensive care units include lack of bonding time, inability to participate in care, and adverse breastfeeding practices [12]. Thus, hospitals are currently struggling to cope with COVID-19 visiting restrictions by changing or adding duties; however, they continue to seek ways to support patients and families.

Visiting restrictions have resulted in changes in the usual and new duties of nurses. Nurses and nursing assistants comprise the largest group of healthcare providers [13] and spend the most amount of time with patients. Since the beginning of the COVID-19 outbreak in 2019, nurses, nursing assistants, and other health professionals have risked their lives to perform their duties [14]. Nurses are currently experiencing high workloads because of the pandemic, and are experiencing physical and emotional stress and moral distress [15]. It has been suggested that fear of COVID-19 not only reduces job satisfaction in frontline nurses but also increases their motivation to leave their jobs [16]. Fear of COVID-19, psychological distress, and job satisfaction are associated with turnover intention among frontline nurses [16]. Psychological distress and stress caused by the COVID-19 pandemic have been widely reported as affecting nurses’ turnover intentions; therefore, this issue must be urgently addressed in frontline nurses [17,18,19,20].

It is important to identify the cause of this problem and address it early before nurses express an intention to leave a facility or the nursing profession. One study suggested that burnout affects turnover intention, but not internal transfer intention [21]. However, there are no other reports on internal transfer intentions during the pandemic, and further investigation is needed. To reduce the psychological burden, pressure, and distress of frontline nurses during the pandemic, early health interventions for nurses are recommended [22]. Therefore, it is important to obtain information about nurses’ intentions regarding internal transfer to prevent turnover. COVID-19 is yet to be eradicated, and efforts are being made to identify ways to live with the virus. Current treatments and vaccinations cannot bring back the pre-pandemic state, and steps must continue to be taken to vaccinate and prevent transmission [13]. The risk of serious illness in hospitalized patients or those with underlying diseases is high [23]; thus, visiting restrictions in hospitals may continue in the future. Identifying workload changes and associated burdens, and changes in the daily life of nurses under family visitation restrictions, would help to reduce physical and mental strain on nurses during the COVID-19 pandemic. By identifying factors related to nurses’ internal transfer intentions, nurse managers may be able to address such factors early. Therefore, this study investigated changes in the work burden and daily life of nurses owing to COVID-19 pandemic-related restrictions on family visits. In addition, it identified the factors that affect nurses’ internal transfer intentions.

## 2. Materials and Methods

### 2.1. Terms Used in This Study 

#### Internal Transfer Intention 

Following Kovner et al. [24], we defined internal transfer intention as nurses’ intention to transfer within the hospital environment. Previous studies have pointed out that definitions of turnover are inconsistent [25,26]. For example, it is important to differentiate between voluntary turnover and involuntary turnover, and between intention to quit hospital work or to quit nursing completely. In this study, internal transfer intention was defined as the intention to transfer within the hospital environment. Because turnover does not include internal transfer, we therefore considered internal transfer to be different from turnover intention.

### 2.2. Data Collection

We partnered with Kanazawa Medical University Hospital in Ishikawa, Japan, to conduct a web-based cross-sectional survey of 625 nurses from one designated medical institution of infectious diseases that accepts COVID-19 patients and two other medical institutions. Designated medical institutions for infectious diseases are institutions that admit patients with new infectious diseases or diseases such as H1N1 influenza, as defined in the Act on the Prevention of Infectious Diseases and Medical Care for Patients with Infectious Diseases (the Infectious Diseases Control Law) of Japan [2]. For the survey, we developed an anonymous self-report questionnaire that assessed changes in workload owing to family visiting restrictions. The questionnaire development was based on interviews with four nurses and studies by Matsuo et al. of St Luke’s International Hospital in Japan [25] and Awano et al. of the Japanese Red Cross Medical Center [26]. A web-based survey was generated using SurveyMonkey, which is a cloud-based survey development application. The study was conducted from 19 November 2021 to 4 March 2022.

### 2.3. Survey Contents

#### 2.3.1. Nurses’ Background

The participants were asked about their age (20s, 30s, 40s, 50s, and 60s), gender, and years of experience.

#### 2.3.2. Workload Changes Owing to Family Visiting Restrictions

Participants were asked to rate whether they found the following tasks burdensome: answering phone calls from family members, providing information about daily life in the hospital, explaining the patient’s condition to family members, looking after valuables such as insurance cards and cash, providing discharge instructions to family members in person, providing discharge instructions to family members by phone, handling complaints from family members, taking care of daily necessities brought for the patient by the family, taking care of money when shopping on the patient’s behalf, shopping for hospital necessities on the patient’s behalf, and videotaping the patient and sending the videos to family members. The response options to these items were “strongly agree”, “agree”, “disagree”, or “strongly disagree”.

#### 2.3.3. Changes in Duties Owing to Family Visiting Restrictions

Participants were asked to rate changes in their duties using the following items: visitor guidance, assisting with visits using the patient’s smartphone, assisting the physician with online explanations to family members, delegating shopping, delegating phone calls, transporting patients who transferred to another hospital or to a nursing home, and transferring the patient to the hospital entrance upon discharge. The response options to these items were “increased significantly”, “increased”, “no change”, “decreased”, or “decreased significantly”.

#### 2.3.4. Changes in Working Hours, Number of Days Off, and Sleeping Hours

Participants were asked to indicate their working hours, number of days off, and sleeping hours using one of the following options: “increased significantly”, “increased”, “no change”, “decreased”, or “decreased significantly”.

#### 2.3.5. Changes in Daily Life after Visiting Restrictions

Participants were asked about whether they engaged in unhealthy eating, had trouble falling asleep, had experienced increased night awakening, had increased their smoking, and had less time to relax. The response options were “strongly agree”, “agree”, “disagree”, or “strongly disagree”.

#### 2.3.6. Need for Support

Participants were asked to indicate whether they wanted the following types of support to be provided: an increase in the number of nurses, an increase in the number of nursing assistants, to be thanked, to be respected, to have a pay rise, to receive counseling, to have a workload reduction, to receive infection prevention education, and to receive parenting support. The response options were “strongly agree”, “agree”, “disagree”, or “strongly disagree”.

#### 2.3.7. Internal Transfer Intention 

Nurses were asked whether they had internal transfer intentions after the introduction of COVID-19-related family visiting restrictions. The response options were “yes” or “no”.

### 2.4. Analytical Method

Data were analyzed for participants who provided responses to all of the following item categories: background, changes in workload owing to family visiting restrictions, work changes owing to family visitation restrictions, working hours, number of days off, change in sleep duration, change in daily life after the imposition of visiting restrictions, need for support, and intention of internal transfer from their department. Participants were divided into two categories according to age (“20s” and “≥30”) and median number of years of experience (<11 years and ≥11 years).

For workload changes owing to family visiting restrictions, changes in daily life after imposition of visiting restrictions, and need for support, “strongly agree/agree” responses were merged into an “agree” category and “disagree/strongly disagree” responses merged into a “disagree” category. For changes in duties, working hours, number of days off, and sleeping hours owing to family visiting restrictions, “increased significantly/increased” responses were merged into an “increased” category, and other responses into an “others” category. These two categories were used in the analyses. 

The association between the intention for internal departmental transfer and background, changes in workload, changes in working hours, number of days off, sleeping hours, changes in daily life, and the need for support after the imposition of visiting restrictions were examined using the chi-square test or Fisher’s exact test. Binomial logistic regression analysis was used to evaluate the factors associated with internal transfer intention (the dependent variable). We used age, gender, and years of experience as covariates. The following independent variables had a significance of less than 5% in the univariate analysis: videotaping the patient and sending the video to family members, transporting patients who transferred to another hospital or to a nursing home, transferring the patient to the hospital entrance upon discharge, and working hours. In addition, the effect of changes in daily life after the imposition of visitation restrictions and the need for support on internal transfer intention was evaluated; age, gender, and years of experience were forcibly entered into the regression. Univariate analysis resulted in a significance level of less than 5% for the following four variables: I have trouble falling asleep, I have experienced increased night awakening, I have less time to relax, and desire to receive counseling; these variables were used to perform stepwise binomial logistic regression analysis. All variables were entered after checking for multicollinearity (variance inflation factor ≥ 10). The significance level was set at 5%. SPSS Ver27 (IBM Corp.; Armonk, NY, USA) was used for all statistical analyses.

### 2.5. Ethical Considerations

This research was conducted in accordance with the Declaration of Helsinki, 1995 (as revised in Seoul, 2008) and carried out with the consent of the university medical research ethics review committees at the authors’ universities (No. I673). A letter of informed consent was distributed to the participants via email, which informed participants of the purpose and significance of the study, the survey methodology, voluntary nature of participation, anonymity of participants’ responses, and that completion of the questionnaire implied their consent.

## 3. Results

Of the total sample of 625 nurses, 152 responded to the survey. Of these, 84 (55.3%) responded to all the survey items and their data were included in the analyses. A total of 26 (31.0%) nurses were in their 20s; 58 (69.0%) were ≥30 years; 3 (3.6%) were men; and 81 (96.4%) were women. The median (range) years of experience was 11 years (1–40) (Table 1).

In terms of workload changes associated with family visiting restrictions, the items with the most positive responses included shopping for hospital necessities on the patient’s behalf (N = 69; 82.1%), taking care of money when shopping on the patient’s behalf (N = 67; 79.8%), and taking care of daily necessities brought for the patient by the family (N = 66; 78.6%). Regarding changes in duties owing to family visitation restrictions, the items with the most positive responses were assisting the physician with online explanations to family members (N = 69; 82.1%), delegating phone calls (N = 69; 82.1%), and delegating shopping (N = 61; 72.6%). Regarding changes in working hours, number of days off, and sleeping hours, respondents reported increases in working hours (N = 41; 48.8%), number of days off (N = 1; 1.2%), and sleeping hours (N = 2; 2.4%), respectively. Details are shown in Table 1.

Regarding changes in daily life after the imposition of visiting restrictions, the items with the most positive responses were less time to relax (N = 56; 66.7%), increased night awakening (N = 34; 40.5%), and difficulty sleeping (N = 33; 39.3%). In terms of need for support, the items with the most positive responses were wanting an increase in the number of nurses (N = 82; 97.6%), wanting an increase in the number of nursing assistants (N = 82; 97.6%), and wanting to have a workload reduction (N = 81; 96.4%). Details are shown in Table 2.

Regarding internal departmental transfer intention, 49 (58.3%) respondents answered “yes”.

In relation to nurses’ backgrounds, and changes in workload, duties, working hours, number of days off, sleeping hours, daily life, and need for support owing to family visiting restrictions, the proportion of nurses who reported having internal transfer intentions was significantly higher among those who reported that it was burdensome to videotape the patient and send the videos to family members (N = 33, 67.3%) (*p* = 0.047); among those who had experienced an increase in the duties of transporting patients who transferred to another hospital or to a nursing home (N = 19, 76.0%) (*p* = 0.033) and transporting patients to the hospital entrance upon discharge (N = 37, 66.1%) (*p* = 0.042); and among those who reported increased working hours (N = 29, 70.7%) (*p* = 0.024) (Table 1). Regarding changes in daily life after the imposition of visiting restrictions, the proportion of nurses who reported having internal departmental transfer intentions was significantly higher among those who reported having trouble falling asleep (N = 25, 75.8%) (*p* = 0.009), having experienced increased night awakening (N = 25, 73.5%) (*p* = 0.020), and having less time to relax (N = 37, 66.1%) (*p* = 0.042). In terms of needing support, the proportion of nurses who reported having internal transfer intentions was significantly higher among those who desired to receive counseling (N = 13, 81.3%) (*p* = 0.039) (Table 2).

A binomial logistic regression analysis was conducted to examine the effects on internal transfer intentions of changes in workload, duties, working hours, number of days off, and sleeping hours owing to family visitation restrictions. Factors affecting internal transfer intentions were being aged ≥30 years [odds ratio (OR): 6.54, 95% confidence interval (CI): 1.19–35.83], having ≥11 years of experience (OR: 12.57, 95% CI: 2.32–68.02), and increased working hours (OR: 4.51, 95% CI: 1.48–13.72) (Table 3).

The effects of changes in daily life and need for support on internal transfer intentions following the imposition of visiting restrictions were greater among nurses with ≥11 years of experience (OR: 4.31, 95% CI: 1.09–17.04), those who reported having experienced increased night awakening (OR: 3.68, 95% CI: 1.33–10.18), and those who desired to receive counseling (OR: 4.38, 95% CI: 1.07–17.91) (Table 4).

## 4. Discussion

Of the participants in this study, 58.3% expressed an intention to internally transfer from their current department. A pre-pandemic study conducted at a general hospital in Japan reported that 45.6% of nurses expressed internal transfer intentions [27]. Therefore, the percentage of nurses expressing internal transfer intentions was higher in the present study, which examined post-pandemic transfer intention. Falatah has shown that turnover intention has increased since before the pandemic [28]. Therefore, the COVID-19 pandemic may have led to an increase in the number of nurses who desire internal transfer.

Internal departmental transfer intention was significantly higher among nurses who reported that their working hours had increased owing to family visiting restrictions than among those who did not. Previous research has shown that longer working hours are associated with burnout [29], which is consistent with the results of the present study. 

It was not possible to simply compare pre-pandemic working hours with working hours during the pandemic in the present study. However, it has been widely reported that the COVID-19 crisis has increased nurses’ intentions to leave the workforce [20,30]. Within-hospital transfer represents a subset of turnovers; given that the retention of new nurses at the unit level may affect future turnover prevention [24], excess working hours for frontline nurses should be considered an important issue. 

Internal transfer intention was significantly higher for nurses aged ≥30 years than for those in their 20s, and for those with ≥11 years of experience. This indicates that nurses who are mid-career or later are more likely to express internal transfer intentions if they are asked to do additional unskilled work because they feel they are overqualified for such work. It is necessary to determine the reasons for increases in working hours and take counter measures as soon as possible. The identification of relevant factors in the present study could help to determine appropriate measures and retain nurses.

Internal transfer intention was significantly higher among nurses who reported increased night awakening and desired to receive counseling than among those who did not. In wards with pandemic-related family visitation restrictions, increased night awakening and a desire for counseling may be important predictors of nurses’ internal transfer intentions. Previous reports have shown that healthcare professionals who treat COVID-19 patients have developed psychological distress, including anxiety and depression [31,32,33]. Factors that affect sleep include distress from insomnia and worsening sleep conditions [34,35,36]. The importance of psychological health interventions for frontline nurses during the COVID-19 pandemic has been noted [22]. It is widely recognized that many hospitals provide a relaxed atmosphere in which nurses can freely discuss their concerns, and provide professional psychological counseling when needed [37,38]. Nurses who express a desire for counseling may be experiencing emotional distress or stress. However, even though internal transfer intention does not always lead to turnover intention, and various factors affect this process, paying attention to increased night awakening and the need for counseling may prevent future turnover.

## 5. Limitations

This study had several limitations. The hospitals to which participants belonged were located in a limited area, the type of hospital was limited to general hospitals, and the response rate was low (24.3%). Data were analyzed for 84 participants (approximately 60% of the respondents); however, this represented only 13.4% of the 625 nurses surveyed, which may reduce the generalizability of the findings. It is possible that many respondents had the intention to transfer internally from their current department and wanted to make their situation known. Conversely, many nurses may have had the intention to transfer from their current department but felt unable to express their intention to transfer because of the pandemic and so did not respond to the survey. It is possible that some nurses reported that they had no internal transfer intentions because they had an external transfer intention. Because this was a cross-sectional study, it was not possible to establish a causal relationship between the variables under investigation. Future longitudinal studies are needed to identify how the causes of nurses’ internal transfer intentions relate to COVID-19-related family visiting restrictions.

## 6. Conclusions

Factors influencing internal transfer intentions were age ≥ 30 years (OR: 6.54, 95% CI: 1.19–35.83); ≥11 years of experience (OR: 12.57, 95% CI: 2.32–68.02); and longer working hours (OR: 4.51, 95% CI: 1.48–13.72). The effect of visitation restrictions on daily life and internal transfer intentions was greater in nurses with ≥11 years of experience (OR: 4.31, 95% CI: 1.09–17.04), those with increased night awakening (OR: 3.68, 95% CI: 1.33–10.18), and those who desired to receive counseling (OR: 4.38, 95% CI: 1.07–17.91).

On the basis of the present findings, we recommend that attention be paid to factors that affect internal transfer intentions of nurses before they develop into turnover intentions. Specifically, it is important to pay more attention to the excess hours worked by nurses during the COVID-19 pandemic, especially those in their 30s and those with more than 11 years of experience.

We consider internal transfer intention as one of the factors that indicates a future intention to leave the workforce, and recommend close monitoring of nurses’ sleep patterns and their desire to receive counseling.

## Figures and Tables

**Table 1 healthcare-10-00959-t001:** Internal transfer intention by nurses’ background, workload, duties, working hours, number of days off, and sleeping hours owing to family visiting restrictions.

			Internal Transfer Intention		
Item	Category	Total	Yes	No	
		N (%)	N (%)	N (%)	*p* Value
Nurses’ background					
Age group	20s	26 (31.0)	16 (61.5)	10 (38.5)	0.690 ^(a)^
	≥30 years	58 (69.0)	33 (56.9)	25 (43.1)	
Gender	Men	3 (3.6)	2 (66.7)	1 (33.3)	1.000 ^(b)^
	Women	81 (96.4)	47 (58.0)	34 (42.0)	
Years of experience	<11	43 (51.2)	29 (67.4)	14 (32.6)	0.083 ^(a)^
	≥11	41 (48.8	20 (48.8)	21 (51.2)	
Workload changes owing to family visiting restrictions					
Answering phone calls from family members	Agree	58 (69.0)	36 (62.1)	22 (37.9)	0.300 ^(a)^
	Disagree	26 (31.0)	13 (50.0)	13 (50.0)	
Providing information about daily life in the hospital	Agree	59 (70.2)	33 (55.9)	26 (44.1)	0.493 ^(a)^
	Disagree	25 (29.8)	16 (64.0)	9 (36.0)	
Explaining the patient’s condition to family members	Agree	55 (65.5)	32 (58.2)	23 (41.8)	0.969 ^(a)^
	Disagree	29 (34.5)	17 (58.6)	12 (41.4)	
Ttaking care of valuables such as insurance cards and cash	Agree	65 (77.4)	36 (55.4)	29 (44.6)	0.311 ^(a)^
	Disagree	19 (22.6)	13 (68.4)	6 (31.6)	
Providing discharge instructions to family members in person	Agree	51 (60.7)	28 (54.9)	23 (45.1)	0.428 ^(a)^
	Disagree	33 (39.3)	21 (63.6)	12 (36.4)	
Providing discharge instructions to family members by phone	Agree	52 (61.9)	34 (65.4)	18 (34.6)	0.095 ^(a)^
	Disagree	32 (38.1)	15 (46.9)	17 (53.1)	
Handling complaints from family members	Agree	64 (76.2)	39 (60.9)	25 (39.1)	0.386 ^(a)^
	Disagree	20 (23.8)	10 (50.0)	10 (50.0)	
Taking care of daily necessities brought for the patient by the family	Agree	66 (78.6)	40 (60.6)	26 (39.4)	0.418 ^(a)^
	Disagree	18 (21.4)	9 (50.0)	9 (50.0)	
Taking care of money when shopping on the patient’s behalf	Agree	67 (79.8)	40 (59.7)	27 (40.3)	0.614 ^(a)^
	Disagree	17 (20.2)	9 (52.9)	8 (47.1)	
Shopping for hospital necessities on the patient’s behalf	Agree	69 (82.1)	43 (62.3)	26 (37.7)	0.112 ^(a)^
	Disagree	15 (17.9)	6 (40.0)	9 (60.0)	
Videotaping the patient and sending the videos to family members	Agree	49 (58.3)	33 (67.3)	16 (32.7)	0.047 ^(a)^
	Disagree	35 (41.7)	16 (45.7)	19 (54.3)	
Changes in duties owing to family visiting restrictions					
Visitor guidance	Increased	61 (72.6)	36 (59.0)	25 (41.0)	0.836 ^(a)^
	Others	23 (27.4)	13 (56.5)	10 (43.5)	
Assisting with visits using the patient’s smartphone	Increased	61 (72.6)	34 (55.7)	27 (44.3)	0.432 ^(a)^
	Others	23 (27.4)	15 (65.2)	8 (34.8)	
Assisting the physician with online explanations to family members	Increased	69 (82.1)	40 (58.0)	29 (42.0)	0.885 ^(a)^
	Others	15 (17.9)	9 (60.0)	6 (40.0)	
Delegating shopping	Increased	61 (72.6)	34 (55.7)	27 (44.3)	0.432 ^(a)^
	Others	23 (27.4)	15 (65.2)	8 (34.8)	
Delegating phone calls	Increased	69 (82.1)	40 (58.0)	29 (42.0)	0.885 ^(a)^
	Others	15 (17.9)	9 (60.0)	6 (40.0)	
Transport of patients in transfer to other hospital or nursing home	Increased	25 (29.8)	19 (76.0)	6 (24.0)	0.033 ^(a)^
	Others	59 (70.2)	30 (50.8)	29 (49.2)	
Transferring the patient to the hospital entrance upon discharge	Increased	56 (66.7)	37 (66.1)	19 (33.9)	0.042 ^(a)^
	Others	28 (33.3)	12 (42.9)	16 (57.1)	
Changes in working hours, number of days off, and sleeping hours					
Working hours	Increased	41 (48.8)	29 (70.7)	12 (29.3)	0.024 ^(a)^
	Others	43 (51.2)	20 (46.5)	23 (53.5)	
Number of days off	Increased	1 (1.2)	1 (100.0)	0 (0.0)	1.000 ^(b)^
	Others	83 (98.8)	48 (57.8)	35 (42.2)	
Sleeping hours	Increased	2 (2.4)	1 (50.0)	1 (50.0)	1.000 ^(b)^
	Others	82 (97.6)	48 (58.5)	34 (41.5)	

^(a)^ χ2 test and ^(b)^ Fisher’s exact test.

**Table 2 healthcare-10-00959-t002:** Desire for internal departmental transfer by changes in daily life after imposition of visiting restrictions and by required support.

			Internal Transfer Intention		
Item	Category	Total	Yes	No	
		N (%)	N (%)	N (%)	*p* Value
Changes in daily life after visiting restrictions					
I engage in unhealthy eating	Agree	27 (32.1)	17 (63.0)	10 (37.0)	0.554 ^(a)^
	Disagree	57 (67.9)	32 (56.1)	25 (43.9)	
I have trouble falling asleep	Agree	33 (39.3)	25 (75.8)	8 (24.2)	0.009 ^(a)^
	Disagree	51 (60.7)	24 (47.1)	27 (52.9)	
I have experienced increased night awakening	Agree	34 (40.5)	25 (73.5)	9 (26.5)	0.020 ^(a)^
	Disagree	50 (59.5)	24 (48.0)	26 (52.0)	
I have increased my smoking	Agree	1 (1.2)	0 (0.0)	1 (100.0)	0.417 ^(b)^
	Disagree	83 (98.8)	49 (59.0)	34 (41.0)	
I have less time to relax	Agree	56 (66.7)	37 (66.1)	19 (33.9)	0.042 ^(a)^
	Disagree	28 (33.3)	12 (42.9)	16 (57.1)	
Need for support					
Wants an increase in the number of nurses	Agree	82 (97.6)	48 (58.5)	34 (41.5)	1.000 ^(b)^
	Disagree	2 (2.4)	1 (50.0)	1 (50.0)	
Wants an increase in the number of nursing assistants	Agree	82 (97.6)	47 (57.3)	35 (42.7)	0.508 ^(b)^
	Disagree	2 (2.4)	2 (100.0)	0 (0.0)	
Wants to be thanked	Agree	47 (56.0)	28 (59.6)	19 (40.4)	0.795 ^(a)^
	Disagree	37 (44.0)	21(56.8)	16 (43.2)	
Wants to be respected	Agree	23 (27.4)	13 (56.5)	10 (43.5)	0.836 ^(a)^
	Disagree	61 (72.6)	36 (59.0)	25 (41.0)	
Wants a pay rise	Agree	80 (95.2)	48 (60.0)	32 (40.0)	0.303 ^(b)^
	Disagree	4 (4.8)	1 (25.0)	3 (75.0)	
Desire to receive counseling	Agree	16 (19.0)	13 (81.3)	3 (18.8)	0.039 ^(a)^
	Disagree	68 (81.0)	36 (52.9)	32 (47.1)	
Wants to have a workload reduction	Agree	81 (96.4)	49 (60.5)	32 (39.5)	0.069 ^(b)^
	Disagree	3 (3.6)	0 (0.0)	3 (100.0)	
Wants to receive infection prevention education	Agree	55 (65.5)	35 (63.6)	20 (36.4)	0.175 ^(a)^
	Disagree	29 (34.5)	14 (48.3)	15 (51.7)	
Wants to receive parenting support	Agree	47 (56.0)	28 (59.6)	19 (40.4)	0.795 ^(a)^
	Disagree	37 (44.0)	21 (56.8)	16 (43.2)	

^(a)^ χ2 test and ^(b)^ Fisher’s exact test.

**Table 3 healthcare-10-00959-t003:** Effects of chronological age, years of experience, and working hours on internal transfer intention.

Item	Category	OR	95% CI		*p* Value
			Lower Limit	Upper Limit	
Gender	Female/male	0.38	0.03	5.29	0.47
Chronological age	≥30/20s	6.54	1.19	35.83	0.03
Years of experience	≥11/<11	12.57	2.32	68.02	0.00
Videotaping the patient and sending the video to family members	No burden/burden	2.41	0.82	7.10	0.11
Transport of patients in transfer to other hospital or nursing home	Increased/others	1.77	0.49	6.42	0.39
Transferring the patient to the hospital entrance upon discharge	Increased/others	2.49	0.69	9.03	0.17
Working hours	Increased/others	4.51	1.48	13.72	0.01

Binomial logistic regression analysis. Abbreviations: CI, confidence interval, OR: odds ratio.

**Table 4 healthcare-10-00959-t004:** Effects of years of experience, increased night awakening, and desire to receive counseling on internal transfer intention.

Item	Category	OR	95% CI		*p* Value
			Lower Limit	Upper Limit	
Gender	Female/male	0.48	0.04	6.31	0.58
Chronological age	≥30/20s	2.43	0.57	10.34	0.23
Years of experience	≥11/<11	4.31	1.09	17.04	0.04
I have experienced increased night awakening	Agree/disagree	3.68	1.33	10.18	0.01
Desire to receive counseling	Agree/disagree	4.38	1.07	17.91	0.04

Binomial logistic regression analysis. Inputted variables: difficulty sleeping, increased night awakening, less relaxation time, and desire to receive counseling.

## Data Availability

The data analyzed during this study are included in this published article. Further inquiries can be directed to the corresponding authors.

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
