# Peer review of "Factors Affecting Nurses’ Internal Transfer Intentions after the Introduction of COVID-19-Related Family Visiting Restrictions"

_healthcare, 2022, doi:10.3390/healthcare10050959_

Round 1

Reviewer 1 Report

Overall this manuscript describing nurses' intention to transfer jobs internally was very good and an important consideration for the literature. Before it is published there are some clarifications and edits that need to be made to avoid confusion or misinterpretation of what the authors are trying to say. 

  1. Page 1, line 39: "Lack of cooperation among family members in explaining..." makes it sound like you are saying the family members are being uncooperative with each other. Please reword to make it more clear what point you are trying to make. 
  2. page 2 line 46: recommend changing the wording to "well-being of patients and their families" instead of "well-being of patients' families."
  3. page 2 line 51: when you say "assessing the terminal phase of residents" are you referring to determination of "end of life/ prognosis?" In Japan, do hospital physicians also have to make this determination or does end of life status not affect the visitation policies in the hospital?
  4. Page 2 line 72-73: in my work setting, I would argue that CNA's spend more time with patients than nurses. Recommend the authors soften/clarify the sentence to account for nursing assistants as well. 
  5. I very much appreciate that the authors include discussion about the nursing home setting in the introduction. However I'm unclear whether nursing home nurses were included in this study. From the methods section it reads that they are targeting only hospital nurses. Did the institutions targeted include both hospital and nursing home  environments? On page 4, line 150 there is a comment about "transporting patients to the hospital during transfers." This is confusing, because I thought the patients were already in the hospital. Are you referring to nursing home nurses having to transport patients back to the hospital? 
  6. Is it possible that nurses who stated they did not have internal transfer intentions, might have said "no" because they actually had external transfer intentions?
  7. Page 5, results: It says there are 152 people who responded but only 84 were included in the analysis. Why were 68 people excluded from the analysis? 
  8. Page 9, line 312-314:  recommend rewording the statement "role involves more social contact." My interpretation of your findings is that skilled nurses don't appreciate having to do non-nursing tasks that are below their level of training. So it's not necessarily that they don't enjoy "social contact" with their patients, it's that the extra work that is making them work longer hours, is work for which they are over qualified. 
  9. Page 9-10: I appreciate the attention to a desire to receive counseling. However, the way a need for counseling is discussed in the manuscript makes it sound as if a desire to seek counseling is a bad thing. I am concerned that a person reading the article might come away thinking that the nurses seeking counseling are weaker than the nurses who aren't seeking counseling (for example the statement "nurses who wish to receive counseling...may find it difficult to cope on their own"). Really a desire to seek counseling is a sign of a person being mature enough to seek a healthy way to manage stress. I believe what the authors are trying to say is that the work has become so burdensome that the employee is feeling a need to seek counseling. So the problem is the workplace and not the person. Please reword the discussion to avoid introducing a negative perception of counseling. 
  10. Was there any compensation for filling out the survey? compensation may entice more people to fill it out in the future. 

Author Response

Dear Reviewer: 1

Thank you for your comments. The paper has been revised as follows.

1.    Page 1, line 39: "Lack of cooperation among family members in explaining..." makes it sound like you are saying the family members are being uncooperative with each other. Please reword to make it more clear what point you are trying to make.

L 39–40
We have revised the text as follows:
However, the lack of family support when explaining treatment and care plans to patients,

2.    page 2 line 46: recommend changing the wording to "well-being of patients and their families" instead of "well-being of patients' families."

L 46
We have revised the text as follows:
According to a report on the effects of COVID-19 visiting restrictions, nursing homes have established local protocols for accepting visitors, which have had a positive effect on and contributed to the well-being of patients and their families.

3.    page 2 line 51: when you say "assessing the terminal phase of residents" are you referring to determination of "end of life/ prognosis?" In Japan, do hospital physicians also have to make this determination or does end of life status not affect the visitation policies in the hospital?

L 51
We have revised the text as follows:
but they face the added dilemma of assessing the dying phase of residents and making exceptions to these restrictions accordingly [6].

Doctors in Japanese hospitals do also have to make this decision. Moreover, it affects the hospital visitation policies.

4.    Page 2 line 72-73: in my work setting, I would argue that CNA's spend more time with patients than nurses. Recommend the authors soften/clarify the sentence to account for nursing assistants as well.

L 71, 73
We have revised the text as follows:
Visiting restrictions have resulted in changes in the usual and new duties of nurses. Nurses and nursing assistants comprise the largest group of healthcare providers [13] and spend the most amount of time with patients. Since the beginning of the COVID-19 outbreak in 2019, nurses, nursing assistants and other health professionals have risked their lives to perform their duties [14].

5.    I very much appreciate that the authors include discussion about the nursing home setting in the introduction. However I'm unclear whether nursing home nurses were included in this study. From the methods section it reads that they are targeting only hospital nurses. Did the institutions targeted include both hospital and nursing home  environments? On page 4, line 150 there is a comment about "transporting patients to the hospital during transfers." This is confusing, because I thought the patients were already in the hospital. Are you referring to nursing home nurses having to transport patients back to the hospital?

L 149
Nursing home nurses were not included in the study.
We have clarified this expression as follows:
2.3.3. Changes in duties owing to family visiting restrictions
Participants were asked to rate changes in their duties using the following items: visitor guidance, assisting with visits using the patient's smartphone, assisting the physician with online explanations to family members, delegating shopping, delegating phone calls, transport of patients in transfer to other hospital or nursing home, and transferring the patient to the hospital entrance upon discharge. The response options to these items were “increased significantly,” “increased,” “no change,” “decreased,” or “decreased significantly.”

L 200–201
The following independent variables had a significance of less than 5% in the univariate analysis: videotaping the patient and sending the video to family members, transport of patients in transfer to other hospital or nursing home, transferring the patient to the hospital entrance upon discharge, and working hours. In addition, the effect of changes in daily life after the imposition of visitation restrictions and the need for support on internal transfer intention was evaluated; age, gender, and years of experience were forcibly entered into the regression.

L 257–258
In relation to nurses' backgrounds, and changes in workload, duties, working hours, number of days off, sleeping hours, daily life, and need for support owing to family visiting restrictions, the proportion of nurses who reported having internal transfer intentions was significantly higher among those who reported that it was burdensome to videotaping the patient and sending the videos to family members (N=33, 67.3%) (p=0.047); among those who had experienced an increase in the duties of transporting patients who transferred to another hospital or a nursing home (N=19, 76.0%) (p=0.033) and transporting the patient to the hospital entrance upon discharge (N=37, 66.1%) (p=0.042); and among those who reported increased working hours (N=29, 70.7%) (p=0.024) (Table 1).

Table 1, 3

We have revised the text as follows:
transporting patients to the hospital during transfers.

transport of patients in transfer to other hospital or nursing home

6.    Is it possible that nurses who stated they did not have internal transfer intentions, might have said "no" because they actually had external transfer intentions?

L 344–346
To mention this possibility, we have added the following text:
It is possible that you answered "no" to the internal transfer intentions because you had an external transfer intention.

7.    Page 5, results: It says there are 152 people who responded but only 84 were included in the analysis. Why were 68 people excluded from the analysis?

L 222–223
We have clarified this information as follows:
152 responded to the survey. Of these, 84 (55.3%) responded to all the survey items their date were included in the analyses.

8.    Page 9, line 312-314: recommend rewording the statement "role involves more social contact." My interpretation of your findings is that skilled nurses don't appreciate having to do non-nursing tasks that are below their level of training. So it's not necessarily that they don't enjoy "social contact" with their patients, it's that the extra work that is making them work longer hours, is work for which they are over qualified.

L 313–316
We have revised the text as follows:
This indicates that nurses who are mid-career or later are more likely to express internal transfer intentions if they are asked to do additional unskilled work because they feel they are overqualified for such work.

9.    Page 9-10: I appreciate the attention to a desire to receive counseling. However, the way a need for counseling is discussed in the manuscript makes it sound as if a desire to seek counseling is a bad thing. I am concerned that a person reading the article might come away thinking that the nurses seeking counseling are weaker than the nurses who aren't seeking counseling (for example the statement "nurses who wish to receive counseling...may find it difficult to cope on their own"). Really a desire to seek counseling is a sign of a person being mature enough to seek a healthy way to manage stress. I believe what the authors are trying to say is that the work has become so burdensome that the employee is feeling a need to seek counseling. So the problem is the workplace and not the person. Please reword the discussion to avoid introducing a negative perception of counseling.

L 321–323
We have revised the text as follows:
In wards with pandemic-related family visitation restrictions, increased night awakening and a desire for counseling may be important predictors of nurses' internal transfer intentions.

L 333
However, even though internal transfer intention does not always lead to turnover intention, and various factors affect this process, paying attention to increased night awakening and the need for counseling may prevent future turnover.

10.    Was there any compensation for filling out the survey? compensation may entice more people to fill it out in the future.

There was no compensation for filling out in this survey. In the next study, we will consider compensation for filling out.

Reviewer 2 Report

The manuscript is worth of publishing. It is well written. However, some of the comments are provided to make it better.

Title: current title is a bit confusing. Suggested title to reframe “Factors affecting nurses’ internal transfer intentions after the introduction of COVID-19 related family visiting restrictions”

Line 99: This study is quantitative thus suggested to use different terms than clarify.

Line 109-111: It is conflicting writing with the above definition. Is it to leave the organization in this study?

Line 137: Elaborate more “taking care of valuables”

Line 187: Do you mean merged? Please replace the term “collapsed” with “merged” in all the related writings.

Materials and Methods:

Validity and reliability of the instruments used in the study: Missing

Details of data collection procedure: Missing

Discussion: First paragraph (line 293-300) the discussion of intention to internal transfer before pandemic and pandemic situation is different thus suggested to discuss according to the pandemic scenario.

Conclusion: Suggested to rewrite focusing on the objective of the study followed by recommendation

Reference no. 5: Check the guideline

Author Response

Dear Reviewer: 2

Thank you for your comments. The paper has been revised as follows.

Title: current title is a bit confusing. Suggested title to reframe “Factors affecting nurses’ internal transfer intentions after the introduction of COVID-19 related family visiting restrictions”

We have revised the title as follows:

Factors affecting nurses’ internal transfer intentions after the introduction of COVID-19 related family visiting restrictions

Line 99: This study is quantitative thus suggested to use different terms than clarify.

L 98

We have revised the text as follows:

Therefore, this study investigates changes in the work burden and daily life of nurses owing to COVID-19 pandemic-related restrictions on family visits.

Line 109-111: It is conflicting writing with the above definition. Is it to leave the organization in this study?

L 108–111

We have revised the text as follows:

In this study, internal transfer intention was defined as the intention to transfer within the hospital environment. Because turnover does not include internal transfer, we therefore considered it different from turnover intention.

Line 137: Elaborate more “taking care of valuables”

L 137, Table 1

We have revised the text as follows:

Looking after valuables such as insurance cards and cash

Line 187: Do you mean merged? Please replace the term “collapsed” with “merged” in all the related writings.

L 188, 189, 191

We have revised the text as follows:

For workload changes owing to family visiting restrictions, changes in daily life after imposition of visiting restrictions, and need for support, “strongly agree/agree” responses were merged into an “agree” category and “disagree/strongly disagree” responses merged into a “disagree” category. For changes in duties, working hours, number of days off, and sleeping hours owing to family visiting restrictions, “increased significantly/increased” responses were merged into an “increased” category, and other responses into an “others” category. These two categories were used in the analyses.

Conclusion: Suggested to rewrite focusing on the objective of the study followed by recommendation.

L 351–356

We have added the following text:

Factors influencing internal transfer intentions were age ≥30 years (OR: 6.54, 95% CI: 1.19–35.83); ≥11 years of experience (OR: 12.57, 95% CI: 2.32–68.02); and longer working hours (OR: 4.51, 95% CI: 1.48–13.72). The effect of visitation restrictions on daily life and internal transfer intentions was greater in nurses with ≥11 years of experience (OR: 4.31, 95% CI: 1.09–17.04), those with increased night awakening (OR: 3.68, 95% CI: 1.33–10.18), and those who desired to receive counseling (OR: 4.38, 95% CI: 1.07–17.91).

Reference no. 5: Check the guideline

L 400–401

We have revised the reference as follows:

  1. Wakabayashi, R; Changes in Nursing Practice and Continuing Education Brought About by The COVID-19. J. Nurs. Res. Colloq. Tokyo Womens Med. Univ. 2021, 16, 45–48.

Reviewer 3 Report

The main question addressed by the research is factors affecting nurses’ internal transfer intentions during the COVID-19 pandemic. The topic is relevant and interesting. The paper is well written. The text is clear and easy to read. 

Please, add 

Limitations of the paper and Future Studies and Recommendations in the

Conclusions.

Author Response

Dear Reviewer: 3

Thank you for your comments. The paper has been revised as follows.

Limitations of the paper and Future Studies and Recommendations in the Conclusions.

L 335, 344–349

We have revised the text as follows:

  1. Limitations

This study had several limitations. The hospitals to which participants belonged were located in a limited area, the type of hospital was limited to general hospitals, and the response rate was low (24.3%). Data were analyzed for 84 participants (approximately 60% of the respondents); however, this represented only 13.4% of the 625 nurses surveyed, which may reduce the generalizability of the findings. It is possible that many respondents had the intention to transfer internally from their current department and wanted to make their situation known. Conversely, many nurses may have had the intention to transfer from their current department but felt unable to express their intention to transfer because of the pandemic and so did not respond to the survey. It is possible that some nurses reported that they had no internal transfer intentions because they had an external transfer intention. Because this was a cross-sectional study, it was not possible to establish a causal relationship between the variables under investigation. Future longitudinal studies are needed to identify how the causes of nurses’ internal transfer intentions relate to COVID-19-related family visiting restrictions.

L 362 –364

Conclusions

We consider internal transfer intention as one of the factors that indicates a future intention to leave the workforce, and recommend close monitoring of nurses' sleep patterns and their desire to receive counseling.